# PI3K/Akt/mTOR Signaling Pathway in Blood Malignancies—New Therapeutic Possibilities

**DOI:** 10.3390/cancers15215297

**Published:** 2023-11-05

**Authors:** Wojciech Wiese, Julia Barczuk, Olga Racinska, Natalia Siwecka, Wioletta Rozpedek-Kaminska, Artur Slupianek, Radoslaw Sierpinski, Ireneusz Majsterek

**Affiliations:** 1Department of Clinical Chemistry and Biochemistry, Medical University of Lodz, Mazowiecka 5, 92-215 Lodz, Poland; wojciech.wiese@stud.umed.lodz.pl (W.W.); julia.barczuk@stud.umed.lodz.pl (J.B.); olga.racinska@stud.umed.lodz.pl (O.R.); natalia.siwecka@stud.umed.lodz.pl (N.S.); wioletta.rozpedek@umed.lodz.pl (W.R.-K.); 2Department of Pathology, Fox Chase Cancer Center, Temple University, Philadelphia, PA 19111, USA; aslupian@temple.edu; 3Faculty of Medicine, Cardinal Stefan Wyszyński University, 01-938 Warsaw, Poland; r.sierpinski@uksw.edu.pl

**Keywords:** PI3K/Akt/mTOR pathway, leukemia, PI3K inhibitors, mTOR inhibitors, dual PI3K/mTOR inhibitors, Akt inhibitors

## Abstract

**Simple Summary:**

The PI3K/Akt/mTOR pathway plays a crucial role in cancer, including leukemia. Abnormalities in this pathway drive carcinogenesis by inducing uncontrolled growth, increased survival, and treatment resistance. The abovementioned pathway is also disrupted in various types of leukemia, which makes it a potential therapeutic target for this disease. Current treatment approaches for leukemia are limited and fraught with numerous side effects. This review article aims to summarize recent research data on inhibitors of the PI3K/Akt/mTOR pathway. Inhibition of this pathway may potentially provide improved treatment outcomes for leukemia.

**Abstract:**

Blood malignancies remain a therapeutic challenge despite the development of numerous treatment strategies. The phosphatidylinositol-3 kinase (PI3K)/protein kinase B/mammalian target of rapamycin (PI3K/Akt/mTOR) signaling pathway plays a central role in regulating many cellular functions, including cell cycle, proliferation, quiescence, and longevity. Therefore, dysregulation of this pathway is a characteristic feature of carcinogenesis. Increased activation of PI3K/Akt/mTOR signaling enhances proliferation, growth, and resistance to chemo- and immunotherapy in cancer cells. Overactivation of the pathway has been found in various types of cancer, including acute and chronic leukemia. Inhibitors of the PI3K/Akt/mTOR pathway have been used in leukemia treatment since 2014, and some of them have improved treatment outcomes in clinical trials. Recently, new inhibitors of PI3K/Akt/mTOR signaling have been developed and tested both in preclinical and clinical models. In this review, we outline the role of the PI3K/Akt/mTOR signaling pathway in blood malignancies’ cells and gather information on the inhibitors of this pathway that might provide a novel therapeutic opportunity against leukemia.

## 1. Introduction

Blood malignancies are a heterogeneous group of neoplasms arising due to the disruption of normal hematopoiesis [1,2]. They are among the most common cancer types, accounting for 6.5% of all cancers around the world [3]. Hematologic malignancies are generally classified into leukemias, multiple myeloma (MM), non-Hodgkin lymphomas (NHLs), and Hodgkin lymphoma (HL) [4]. Each disease is characterized by different morphological features, prognosis, and treatment regimens [5]. In this review, we focus on four types of hematologic malignancies—acute lymphocytic leukemia (ALL), chronic lymphocytic leukemia (CLL), acute myeloid leukemia (AML), and chronic myeloid leukemia (CML). Activation of the phosphoinositide 3-kinase/Akt/mammalian target of rapamycin pathway (PI3K/Akt/mTOR) has been detected in AML, CML, ALL, and CLL cells. 

The PI3K/Akt/mTOR pathway is one of the most important signaling pathways that regulates cell growth and proliferation. PI3K/Akt/mTOR activation is important for leukemogenesis and it is associated with an unfavorable prognosis. The pathway is apparently dysregulated in numerous human malignancies such as respiratory tumors, digestive system tumors, kidney cancer, skin cancer, breast, and ovarian cancer, to name a few [6,7]. The activation of the PI3K/Akt/mTOR axis has been detected in hematologic malignancies, including ALL, CLL, AML, and CML. PI3K/Akt/mTOR activation is important for leukemogenesis and it is associated with unfavorable prognosis. Inhibition of this pathway through specific inhibitors results in reduced leukemic cell proliferation [8,9]. In T-ALL, mutations and deletions leading to the phosphatase and tensin homolog (*PTEN*) loss are the most common reasons for the upregulation of the PI3K/Akt/mTOR signaling, as *PTEN* is a major negative regulator of this pathway. *PTEN* loss contributes to increased cell proliferation and chemoresistance in AML, ALL, and CML [10]. The PI3K/Akt/mTOR pathway is also known to play an important role in CLL, where it promotes autophagy and thus contributes to improved cell survival [11]. Moreover, the PI3K/Akt/mTOR axis is associated with several mutations in hematological malignancies. For instance, *FLT3* mutations in AML promote proliferation through mTOR signaling, whereas BCR-ABL kinase in CML activates the PI3K/Akt/mTOR pathway by binding to the p85 PI3K regulatory subunit [11,12].

For the abovementioned reasons, inhibition of the PI3K/Akt/mTOR pathway may represent a new therapeutic opportunity against leukemia. In this review, we gathered the latest knowledge about the role of the PI3K/Akt/mTOR pathway in blood cancer, pharmacological inhibitors of the pathway, and their application in hematologic malignancies treatment. 

## 2. The PI3K/Akt/mTOR Signaling Pathway

PI3K is a lipid kinase family that phosphorylates inositol’s 3′-OH group in phospholipids on the plasma membrane [13]. Human cells contain three classes of PI3Ks. Class I PI3Ks consist of a catalytic isoform and a regulatory subunit which mediate the kinase activity. This class is divided into two subclasses—IA, activated by receptor tyrosine kinases (RTKs), and IB, activated by G protein-coupled receptors (GPCR) [14,15]. Genes *PIK3CA*, *PIK3CB,* and *PI3KCD* encode the class IA catalytic subunit isoforms p110α, p110β, and p110δ, respectively. These catalytic isoforms can form heterodimers with any of the regulating subunit isoforms, p85α, p55α or p50α (*PIK3R1*), p85β (*PIK3R2*), and p55γ (*PIK3R3*), all of which are p85α splicing variants. *PIK3CG* encodes the class IB p110γ catalytic subunit, which forms heterodimers with regulatory isoforms—p101 or p87, encoded, respectively, by *PIK3R5* and *PIK3R6*. Interestingly, while p110α and p110β are ubiquitously expressed in all tissues, p110δ and p110γ expression seems to be highly restricted to leukocytes and the hematopoietic system [14,16,17,18]. Class II PI3Ks contain only the catalytic subunit. This subunit has three isoforms—PI3K-C2α, PI3K-C2β, and PI3K-C2γ, encoded, respectively, by *PIK3C2A, PIK3C2B,* and *PIK3C2G*. Class III PI3Ks consist of Vps34 (vacuolar protein sorting 34), which is encoded by *PIK3C3* [15,19] (Table 1). 

PI3K phosphorylates phosphatidylinositol 4,5-bisphosphate (PIP2) to phosphatidylinositol 3,4,5-triphosphate (PIP3). PIP3 is the second messenger that enables the interaction of phosphoinositide-dependent kinase 1 (PDK1) and Akt, resulting in Akt phosphorylation. Akt then promotes the phosphorylation of proteins responsible for enhancing cell growth, proliferation, and protein synthesis. Akt activity is maximal when it Is phosphorylated at both sites at Thr308 and Ser473 [15,20].

Tumor suppressors, such as phosphatase and tensin homolog (PTEN) and inositol polyphosphate 4-phosphatase type II (INPP4B), are involved in dephosphorylation of PIP3 to PIP2, therefore suppressing the activity of Akt and its downstream effectors. The Akt/protein kinase B (PKB) has three isoforms—Akt1 (PKBα), Akt2 (PKBβ), and Akt3 (PKBγ), all of which share similar structure and functions [21,22].

Once Akt is phosphorylated, it is able to promote protein synthesis and inhibit proapoptotic proteins such as BCL-2-antagonist of cell death (BAD), forkhead box protein O1 (FoxO1), BCL-2-like protein 11 (BIM), and pro-caspase-9. Akt activation also results in p53 degradation via phosphorylation of mouse double minute 2 homolog (MDM2) [23]. Akt can activate mTOR which is a serine/threonine protein kinase able to interact with different protein molecules by creating two complexes: mTOR complex 1 (mTORC1) and mTOR complex 2 (mTORC2) [20]. mTORC1 function can be modulated by the PI3K/Akt pathway and rapamycin, whereas mTORC2 is sensitive to growth factors and insensitive to rapamycin, unless the exposure to rapamycin is prolonged [24,25]. Contrary to mTORC1, mTORC2 possesses the ability to regulate Akt activity upon PDK1 direct signal transduction [15]. It is worth mentioning that the mTORC1 signaling pathway is better characterized than mTORC2 [26]. Upon activation, mTORC1 stimulates cell growth and modulates protein synthesis by means of eukaryotic translation initiation factor 4E-binding protein 1 (4E-BP1), ribosomal protein S6 kinase 1 (p70S6K or S6K1), forkhead/winged helix box class O (FOXO) family, RAS/ERK, and many other pathways. On the other hand, mTORC2, by phosphorylating Akt, can control signaling of several growth factors [20,25]. Furthermore, phosphorylation of protein kinase C (PKC)δ, PKCζ, PCKγ, and PKCε by mTORC2 can also promote cytoskeleton construction [26] (Figure 1). 

## 3. The Role of PI3K/Akt/mTOR Pathway in Cancer Cells

The PI3K/Akt/mTOR pathway plays a crucial role in various hallmarks of cancer; these include sustaining proliferative signaling, evading growth suppressors, activating invasion and metastasis, and deregulating cellular energetics [27]. Moreover, it is known to promote resistance to commonly used therapeutic methods like chemotherapy and immunotherapy, which, therefore, leads to disease progression [28].

Overactivation of the PI3K/Akt/mTOR axis in tumor cells can be caused either by genetic mutations or, more frequently, by post-translational modifications. One of the most important epigenetic regulators of the pathway constitutes of small noncoding RNAs called microRNAs (miRNAs) [29]. The most common alterations in the PI3K/Akt/mTOR pathway found in human cancers include activating mutations in *PIK3CA*, loss of function mutations and deletions in *PTEN*, amplification and activation of specific PI3K-activating receptor tyrosine kinases like EGFR and HER2, and amplification and gain-of-function mutations in *AKT1*, *AKT2,* or *AKT3* [30]. Mutation in the *PIK3CA* oncogene is connected with inhibition of apoptosis and acquirement of chemoresistance in triple-negative breast cancer (TNBC) [31]. The ability of Akt signaling to promote cell survival, growth, and proliferation partially results from altering cellular metabolism. Akt can either cause immediate changes by phosphorylating, thereby regulating the activity of metabolic enzymes, or influence metabolism indirectly, by controlling several transcription factors (e.g., FOXO family members) [32].

### 3.1. Glucose Metabolism 

Akt drives glucose uptake into cells through glucose transporters GLUT1 and GLUT4 due to direct inhibition of thioredoxin-interacting protein (TXNIP) [33]. In addition, Akt phosphorylates and activates enzymes directly or indirectly involved in glycolysis, such as hexokinase 2 (HK2) [34] and 6-phosphofructo-2-kinase/fructose-2,6-biphosphatase (PFKFB2) [35]. The phosphorylation of HK2 occurs as a result of Akt/mTORC2 signaling activation [36]. Phosphorylated HK2 binds to the outer mitochondrial membrane and exerts antiapoptotic function [37]. Akt-mediated overactivation of HK2 has been found to correlate with tumor progression and unfavorable prognosis in various malignancies, such as hepatocellular carcinoma and gastric and colorectal cancer [38].

FOXO family members, hypoxia-inducible factor 1 (HIF1), and Myc are among the main Akt-dependent downstream transcription factors engaged in glucose metabolism. HIF1 and Myc regulate glucose uptake and glycolysis by inducing expression of most glycolytic enzymes and major glucose transporters, mainly GLUT1, which plays a crucial role in glucose uptake into cancer cells. Moreover, the inhibition of FOXO through the PI3K/Akt axis contributes to the same effect, as FOXO is known to suppress Myc function [39]. The abovementioned mechanisms occur both physiologically and in cancer pathology. Under conditions of pathological PI3K/Akt signaling in cancer cells, a constitutive induction of the Warburg effect occurs, which enables the synthesis of cellular macromolecules. Both mTOR complexes show the ability to induce this phenomenon: mTORC1 by regulating glucose metabolism and mTORC2 by promoting the expression of glycolytic enzymes [40].

### 3.2. Lipid Synthesis

The PI3K/Akt/mTOR pathway stimulates anabolic processes in tumor cells. For instance, Akt initiates de novo lipid synthesis by direct phosphorylation and, thus, activation of ATP-citrate lyase (ACLY). ACLY is an enzyme which promotes the production of cytosolic acetyl-CoA, a precursor of sterols and fatty acids [41,42]. Furthermore, Akt-mediated ACLY activity increases histone-acetylation levels in cancer cells, particularly in glucose-limited conditions [43]. Another mechanism by which Akt signaling influences lipid synthesis is the activation of the sterol regulatory element binding proteins (SREBPs) family of transcription factors. SREBPs stimulate the expression of enzymes involved in fatty acids and sterols synthesis, which is necessary for building membranes and production of second messengers [44]. Among other mechanisms, Akt can increase SREBPs activity via mTORC1. Moreover, Akt inhibits glycogen synthase kinase 3 (GSK3) by phosphorylating its autoinhibitory N-terminal serine (Ser9/Ser21) [45], which in turn promotes the stability of processed, active forms of SREBPs [46].

### 3.3. Nucleotide Synthesis

Nucleotides, which are required for the synthesis of nucleic acids (RNA and DNA), play a major role in cell growth and proliferation [47]. Thus, in cancer cells, de novo synthesis of purines and pyrimidines is upregulated through various mechanisms, including aberrations in PI3K/Akt/mTOR signaling. The PI3K/Akt/mTOR axis is responsible for glycolytic carbon flux into both oxidative and nonoxidative branches of the pentose phosphate pathway (PPP), which enables the generation of ribose for nucleotide synthesis [48]. Akt directly enhances carbon flow through the nonoxidative PPP by phosphorylating transketolase, a key enzyme for this pathway. As a result, purine synthesis is increased [49]. Akt, via mTORC1, promotes oxidative PPP flux by activating SREBPs, which induces the expression of glucose 6-phosphate dehydrogenase (G6PD) [50]. PPP-related enzymes, including transketolase (TKT) and G6PD, are overexpressed in several malignancies like breast, lung, ovarian, and colorectal cancer, wherein they are known to promote the development of chemoresistance [51]. Another way in which Akt signaling controls nucleotide synthesis is the regulation of Myc activity, as Myc increases the expression of metabolite precursors and many enzymes involved in purine and pyrimidine synthesis [52]. The activation of mTOR downstream branch of PI3K/Akt signaling regulates nucleotide synthesis de novo on both the transcriptional and posttranslational levels. mTORC1 directly influences pyrimidine synthesis by activating the first enzyme in their biosynthetic pathway (CAD—carbamoyl-phosphate synthetase 2, aspartate transcarbamoylase, dihydroorotase) by S6K1-mediated phosphorylation [53]. Meanwhile, purine synthesis is driven by mTORC1 through transcriptional mechanisms involving Myc, SREBPs, and activating transcription factor 4 (ATF4) [54]. 

### 3.4. Protein Synthesis and Degradation

Enhanced protein synthesis is one of the key mechanisms enabling the growth of tumor cells. As a downstream effector of the PI3K/Akt pathway, mTOR plays a major role in regulating protein synthesis and degradation in cancer cells. mTORC1 promotes the cap-dependent initiation of translation by phosphorylating 4E-BP1 [55]. Other crucial targets of mTORC1 are ribosomal protein S6Ks: S6K1 and S6K2. S6Ks, phosphorylated and activated by mTORC1, modulate the functions of translation initiation factors, as well as contribute to cell growth [56]. 

Apart from enhancing protein synthesis, mTORC1 prevents protein degradation through autophagy inhibition [57]. mTORC1 has been found to phosphorylate mammalian autophagy-initiating kinase Ulk1 at Ser757 [58] and transcription factor EB (TFEB) at Ser142 and Ser211 [59], leading to their inactivation. Furthermore, mTORC1 controls autophagy by phosphorylating, and thus inactivating, several autophagy-related (Atg) proteins, such as mammalian Atg13 [60]. mTORC1-mediated autophagy regulation plays a vital role in cancer biology. Autophagy inhibition leads to accumulation of reactive oxygen species (ROS) which increases DNA damage and therefore promotes carcinogenesis [61]. Another mechanism by which mTORC1 controls protein degradation is influencing the ubiquitin proteasome system (UPS) [62]. Nevertheless, despite numerous studies, more data are needed to establish the exact biochemical connection between mTOR and proteasomal protein degradation [63]. 

## 4. Application of PI3K/Akt/mTOR Pathway Inhibitors in Blood Malignancies

Depending on the mechanism of action, PI3K/Akt/mTOR pathway inhibitors are divided into PI3K kinase inhibitors, Akt kinase inhibitors, and mTOR kinase inhibitors. There are also dual inhibitors that inhibit both PI3K and mTOR kinase. Nowadays, the Food and Drug Administration (FDA) has approved two inhibitors of the PI3K/Akt/mTOR pathway for leukemia treatment; these are Idelalisib and Duvelisib, for the treatment of CLL. We gathered all PI3K/Akt/mTOR pathway inhibitors in preclinical and clinical trials and approved by the FDA for leukemia treatment in Table 2.

### 4.1. PI3K Inhibitors

PI3K inhibitors are classified as isoform-specific, pan-PI3K (targeting all four isoforms, α, β, δ, and γ of class I PI3K), and dual PI3K/mTOR inhibitors. Among all pan-PI3K inhibitors, only Copanlisib has been approved by the FDA. The other known pan-PI3K inhibitors are Buparlisib and ZSTK474. Idelalisib and Umbralisib inhibit the δ isoform, whereas Duvelisib inhibits the δ/γ isoforms of PI3K [64]. 

Idelalisib is a PI3Kδ inhibitor approved in 2014 by the FDA. It is used for treating CLL in combination with Rituximab in disease resistance or relapse after at least one line of therapy. Idelalisib is the first-line agent for patients with del17p or TP53 mutations ineligible for immunochemotherapy. Idelalisib is also approved in follicular lymphoma (FL) after two lines of therapy [65]. Despite the FDA approval of Idelalisib for treating CLL and FL, studies on the efficacy of this drug combined with other drugs are constantly underway. The study with Idelalisib in combination with Rituximab presents increased PFS (progression-free survival) in patients with relapsed CLL, compared to placebo plus Rituximab administration. Median PFS in the Idelalisib plus Rituximab (I–R) group was 20.3 months after a median follow-up time of 18 months. Patients who received I–R in the main study and continued treatment with Idelalisib alone had a median PFS of 20.3 months (95% CI, 17.3–26.3 months), and an overall response rate (ORR) of 85.5%. Median OS (overall survival) was 40.6 months and 34.6 months for patients assigned randomly to I–R and placebo plus Rituximab groups, respectively. Treatment with Idelalisib did not increase the incidence of elevated liver aminotransferases; however, it increased the incidence of diarrhea, colitis, and pneumonia [66]. On the other hand, another study revealed that treatment with I–R proved less effective for patients with relapsed/refractory CLL than monotherapy with Acalabrutinib—a Bruton’s tyrosine kinase inhibitor. Median PFS and estimated 12-month PFS were significantly longer in the case of Acalabrutinib monotherapy compared with I–R or Bendamustine plus Rituximab (B–R) therapy. Serious adverse events (Aes) occurred in 29% of patients treated with Acalabrutinib monotherapy, 56% treated with I–R, and 26% treated with B–R. Deaths occurred in 10%, 11%, and 14% of patients receiving Acalabrutinib monotherapy, I–R, and B–R, respectively [67]. Altogether, it has been concluded that I–R therapy is less effective and has more side effects in relapsed/refractory CLL than Acalabrutinib. 

Duvelisib is an orally available dual inhibitor of PI3K-δ approved by the FDA for use in relapsed/refractory CLL or small lymphocytic lymphoma (SLL) after at least two prior therapies [68]. In a Phase I study, Duvelisib monotherapy in patients with relapsed/refractory CLL resulted in an ORR of 56% [69]. Studies on the efficacy of Duvelisib in combination with other drugs are ongoing. One study analyzed the efficacy and toxicity of Duvelisib in combination with Rituximab (ARM 1) and Duvelisib with Bendamustine and Rituximab (ARM 2) in patients with relapsed/refractory non-Hodgkin lymphoma (NHL) and CLL. Patients with CLL had better treatment outcomes in ARM 1 (ARM 1—partial response (PR) 88.9%, ORR 88.9% vs. ARM 2—PR 50%, ORR 75.0%). Duvelisib in combination with Rituximab or Bendamustine and Rituximab did not increase treatment toxicity [70]. The combination of Duvelisib with Rituximab may become a new, promising treatment strategy. However, more studies are needed to confirm their combined efficacy.

The other Inhibitor of pan-PI3K is Buparlisib. Buparlisib therapy in patients with refractory/relapsed CLL resulted in an ORR of 46% with a median duration of response of 15.5 months. The most common side effects after Buparlisib administration were hyperglycemia, fatigue, anxiety, and gastrointestinal toxicities [71,72]. However, in a Phase I trial with patients with refractory/relapsed AML and refractory/relapsed ALL, Buparlisib was less effective, with a median survival time of 75 days [72]. On the other hand, Buparlisib was more cytotoxic against B-CLL cells than Idelalisib in in vitro studies [73].

The next studied PI3K inhibitor is Umbralisib. Umbralisib inhibits PI3Kδ isoform as well as casein kinase-1ε. Patients with CLL have reached a PFS of 23.5 months with Umbralisib therapy [74]. Therapy based on Umbralisib plus Ibrutinib in patients with relapsed/refractory CLL achieved an ORR of 90%, and PR/PR with lymphocytosis accounted for 29% [75]. Every AE was graded according to the Common Terminology Criteria for Adverse Events grade. The most common Aes were diarrhea, nausea, and fatigue, and the most common grade 3 or higher AEs were anemia and thrombocytopenia. The most common AEs leading to early discontinuation of therapy were rash, arthralgia, and atrial fibrillation [74,76,77]. 

Another pan-PI3K inhibitor applied in preclinical studies is ZSTK474. ZSTK474 inhibited the cell growth of AML and ALL in in vitro studies [78,79]. The inhibition of cell growth was concentration-dependent. ZSTK474 monotherapy induced apoptosis both in AML and ALL cells. Upon combination of ZSTK474 with the extracellular signal-regulated kinase 1/2 (ERK1/2) inhibitor AZD0364, the apoptosis of ALL and AML cells was increased; this was associated with the induction of oxidative stress and cellular antioxidant defense mechanisms [79]. Moreover, ZSTK474 exhibited cytotoxic effects against T-ALL and B-ALL cells [80,81]. In a separate study, ZSTK474 reduced CML cells viability and proliferation by inducing cell cycle arrest at the G1 phase. Further, the combination of ZSTK474 with Imatinib showed a synergistic effect, improving the effectiveness of therapy also in the multidrug-resistant counterpart cells [82]. Another in vitro study tested the efficacy of ZSTK474 in combination with Imatinib, Nilotinib, and the BCR-ABL inhibitor GZD824 on Philadelphia chromosome-positive B-ALL cells. The combination of these drugs decreased cell viability and induced apoptosis and autophagy [83]. 

### 4.2. mTOR Inhibitors

The next group of drugs described herein is mTOR inhibitors, already used in immunosuppression and cancer treatment. The first generation of mTOR inhibitors comprises natural rapamycin (Sirolimus) and its synthetic analogs, known as rapalogists. These inhibitors bind to the FKBP-rapamycin-binding (FRB) domain. Second-generation ATP-competitive mTOR inhibitors (TOR-Ki) can effectively block both mTORC1 and mTORC2 by binding to the ATP-binding pocket of the kinase catalytic domain (KIN). The third generation of mTOR inhibitors is called RapaLinks or bi-steric mTORC1 inhibitors; these are made by connecting rapamycin and TOR-Ki. This generation embraces the action of both first- and second-generation mTOR inhibitors [84,85].

Everolimus is an oral mTOR kinase inhibitor used to prevent the rejection of transplanted organs and treat breast cancer, pancreatic-derived neuroendocrine tumors, and renal cell carcinoma, among others [86]. In a randomized trial in patients with AML, Everolimus did not increase relapse-free survival, the cumulative incidence of relapse, or OS. The study randomized patients to receive Everolimus between consolidation chemotherapy courses. The study terminated due to excess mortality in the Everolimus arm, without any evidence of beneficial disease control [87]. In a study of childhood patients with relapsed ALL, therapy with Everolimus combined with Vincristine, Prednisone, Pegaspargase, and Doxorubicin led to complete remission in 19 of 22 consecutive patients. Complete remission occurred in all six patients with a known *KMT2A* or iAMP21 rearrangement. The combination of Everolimus with the drugs mentioned above was well tolerated [88]. Everolimus combined with HyperCVAD (Cyclophosphamide + Vincristine + Doxorubicin + Dexamethasone) chemotherapy resulted in better treatment outcomes than first-line HyperCVAD alone—partial response (PR) or complete response (CR) of 63.6% and 53.3%, respectively. The results were not statistically significant, however, and additional studies on a larger group of patients are needed to confirm the effectiveness of the therapy. The therapy of Everolimus plus HyperCVAD was also well tolerated [89].

Rapamycin (Sirolimus—trade name) is an mTOR inhibitor that was approved by the FDA in 1999 for the prevention of kidney transplant rejection. Rapamycin is a macrocyclic lactone produced by *Streptomyces hygroscopicus*, which was isolated from soil samples in the late 1960s. Rapamycin or its rapalogues are also applied in the prevention of restenosis after coronary angioplasty and used in oncology—the FDA approved the use of rapamycin to treat patients with pancreatic cancer in 2011 [90,91]. In a clinical trial in high-risk AML patients treated with Sirolimus in combination with MEC (Mitoxantrone + Etoposide + Cytarabine), the ORR was 47% (CR 33%, complete remission with incomplete hematologic recovery 2%, PR 12%). Moreover, ORR was not significantly different between participants with and without baseline mTORC1 activity (52% vs. 40%, respectively). Sirolimus therapy together with MEC was well tolerated [92]. In another Phase II study in patients with relapsed/refractory AML, Sirolimus in combination with MEC had an ORR of 16%. The study also examined the efficacy of the Carboplatin + Topotecan and Alvocidib + Cytarabine + Mitoxantrone scheme. The ORR accounted for 14% and 28%, respectively [93].

Temsirolimus is an mTOR inhibitor that was approved by the FDA in 2007 for the treatment of advanced renal cell carcinoma [84]. The combination of Temsirolimus with a UKALL R3 reinduction chemotherapy regimen (Dexamethasone + Vincristine + Mitoxantrone + Pegaspargase + intrathecal Methotrexate) was investigated in childhood relapsed/refractory ALL patients. In the study, 46.6% of patients achieved remission, while 20% had residual disease of <0.01%. In therapy-related AEs, 73% of the children studied developed neutropenic fever and 53% of patients had a documented grade 3 or 4 infection and one grade 5 bacterial sepsis [94]. Another study examined the therapy of Temsirolimus combined with Clofarabine in patients with AML. The treatment resulted in an ORR of 21%, of which 8% achieved CR. Median disease-free survival was 3.5 months, and median OS was 4 months [95]. 

RMC-4627 is a novel bi-steric mTORC1-selective inhibitor; at this moment it is undergoing preclinical studies. In in vitro research, RMC-4627 demonstrated strong and selective inhibition of 4E-BP1 phosphorylation specifically within B-ALL cell lines, while mTORC2 activity was unaffected. RMC-4627 reduced proliferation, decreased survival, and significantly increased the efficacy and tolerability of Dasatinib in a Ph+ B-ALL xenograft model [85,96]. It is worth adding that the first clinical candidate in the class of bi-steric mTORC1 inhibitor (RMC-5552) is undergoing clinical trials in solid tumors (NCT04774952) [97].

### 4.3. Dual PI3K/mTOR Inhibitors

Dual PI3K/mTOR inhibitors are another group of potential drugs undergoing extensive investigation. Dual PI3K/mTOR inhibitors can completely suppress the aberrant activation of the PI3K/Akt/mTOR signaling pathway and prevent the compensatory activation of the Akt/mTOR pathway; this can lead to improved treatment outcomes [98]. 

Gedatolisib was found to reduce the number of ALL cells in the spleen by an average of 91.8% in patient-derived xenograft mouse models of childhood Ph-like ALL. In a xenograft mice with cytokine receptor 2 *(CRLF2)/JAK* Ph-like ALL mutations, Gedatolisib therapy reduced the viability of ALL cells by 92.2%. Gedatolisib also inhibited ALL proliferation in *ABL*/platelet-derived growth factor receptor (*PDGFR*) mutant models with a mean reduction of 66.9%. The high efficacy of Gedatolisib correlated with inhibition of phosphorylated ribosomal protein S6 (pS6) and 4E-BP1 in Ph-like ALL models [99]. Gedatolisib significantly prolonged survival of mice in a xenograft model of Sorafenib-resistant AML [100]. Furthermore, Gedatolisib treatment led to marked inhibition of T-ALL growth compared to vehicle treatment, as well as delayed tumor growth in all treated mice [101]. 

Imidazoquinoline derivative BEZ235 is a dual PI3K/mTOR inhibitor. BEZ235 is currently undergoing Phase I clinical trials. In one of the studies, in patients with refractory or relapsed leukemia, the response was observed in 2 of 10 patients with BCP-ALL and 1 of 1 patient with T-ALL. In contrast, there was no response in any patient with AML (*n* = 12) or CML (*n* = 1). The response to BEZ235 treatment was uncorrelated with the level of PI3K signaling markers. BEZ235 therapy was mainly associated with gastrointestinal-related toxicity [102]. In vitro, BEZ235 reduced viability, induced G0/G1 arrest, and increased apoptosis of AML cells [103,104]. In xenograft models of AML MLL-AF9+/FLT3-ITD+, BEZ235 therapy resulted in delayed tumor progression and prolonged survival [105]. It was also found that BEZ235 inhibited AML cell migration and sensitized cells to Vincristine and Adriamycin [104]. Furthermore, in either in vitro or in vivo models of T-ALL, inhibition of PI3K/mTOR with BEZ235 enhanced the antileukemic effect of Dexamethasone [106]. BEZ235 also inhibited proliferation, induced apoptosis, and activated autophagy in CML cells in both cellular and xenograft models [107,108]. Moreover, BEZ235 increased the sensitivity of CML cells to Imatinib [108]. Another study showed that BEZ235 inhibited proliferation of adult T-cell leukemia (ATL) cells in in vivo models. In CML cells treated with BEZ235, PI3K/Akt/mTOR activity and the levels of the antiapoptotic protein Bcl-2 decreased, while the levels of the proapoptotic protein Bax increased [103,109]. BEZ235 also offers the opportunity to overcome resistance to Venetoclax, a selective Bcl-2 inhibitor. Venetoclax has significantly enhanced the treatment options available to patients with refractory and relapsed blood cancers, including those with AML. Venetoclax has fewer side effects, which makes it more effective for elderly patients. Numerous studies have demonstrated that the two major antiapoptotic Bcl-2 family proteins, namely, Bcl-XL and MCL-1, serve as the primary factors that determine resistance to Venetoclax. Venetoclax has a high binding specificity for Bcl-2; thus, the relative expression levels of Bcl-2 proteins may be a determinant of Venetoclax resistance. Combining Venetoclax with other targeted drugs such as BEZ235, a dual PI3K/mTOR inhibitor, offers a chance to circumvent resistance to Venetoclax [9,11,110,111].

### 4.4. Akt Inhibitors

GSK2141795 is an ATP-competitive, fully reversible pan-Akt kinase inhibitor. GSK2141795 inhibited neoplastic cell proliferation with activated Akt pathway in vitro and in vivo [112]. A Phase II study enrolled patients with relapsed/refractory AML with an *RAS* mutation, in which GSK2141795 therapy was combined with Trametinib (GSK1120212)—a dual-specificity mitogen-activated protein kinase kinase 1 and 2 inhibitor (MEK1 and MEK2). No patient achieved CR and CR with incomplete recovery of platelets due to therapy, and the study was closed early due to lack of clinical activity. The median OS amounted to 3 months. The most common AEs were diarrhea, maculopapular rash, and mucositis. Serious AEs (grade 3–4) were observed in 39% of patients, the most frequent being rash, mucositis, and diarrhea [113].
cancers-15-05297-t002_Table 2Table 2The current drug developmental stages of the specific inhibitors of the phosphoinositide 3-kinase/Akt/mammalian target of rapamycin (PI3K/Akt/mTOR) signaling pathway in preclinical and clinical studies.TargetCompound NameTypeCurrent StatusNumber of Evaluable Patients in TrialResults
IdelalisibPI3K inhibitorFDA-approved 220FDA-approved Idelalisib in patients with relapsed CLL based on a significant PFS benefit. In relapsed follicular lymphoma (FL) and relapsed SLL, Idelalisib was approved under the accelerated approval program based on the tumor objective response rate data [114].
DuvelisibPI3K inhibitorFDA-approved319 Approved by the FDA for relapsed CLL/SLL and FL. Patients with CLL/SLL treated with Duvelisib had a median PFS of 16.4 months and an overall response rate of 78%. FL patients treated with the drug had an overall response rate of 42% [115].PI3KZSTK474PI3K inhibitorPreclinicalIn vitro research AML [78] and CML cell proliferation decrease. Combination of ZSTK474 and Imatinib indicated synergistic effect on both cell lines [82].BuparlisibPI3K inhibitorCT Phase I14Safe, well tolerated, modest efficacy in advanced AML and ALL [72].CT Phase II12Safe, well tolerated, promising results in relapse or refractory CLL—6/12 achieved PR with a median duration of response of 15.5 months [71].UmbralisibPI3K inhibitor CT Phase I22Safe, well tolerated, promising results in CLL—8/22 achieved CR and 14/22 achieved PR [116].90Safe, well tolerated, promising results in relapsed or refractory CLL. 85% of patients with relapsed or refractory CLL achieved an objective response. 8 assessable patients with high-risk cytogenetic features CLL 6 had a response, of whom 2 had a PR [77].CT Phase I/IB44Safe, well tolerated, promising results in relapsed or refractory CLL. The ORR was 90% [75].CT Phase II51Safe, well tolerated, promising results in CLL. The ORR was 44%—19/48 PR and 2/48 CR [74].28Safe, well tolerated, encouraging response in CLL—52% of patients achieved undetectable minimal residual disease [117].
RMC-4627mTOR inhibitor PreclinicalIn vitro researchIn in vitro cell line models of Ph+ B-ALL, RMC-4627 suppressed cell cycle progression, reduced survival, and enhanced Dasatinib cytotoxicity [96].
mTOR
EverolimusmTOR inhibitorCT Phase I22Safe, well tolerated, promising results in childhood ALL with favorable rates, second PR (86%) and low-end reinduction minimal residual disease (68%) [88].Data unpublishedData unpublished (NCT03328104, NCT03740334, NCT01154439, NCT00819546, NCT00636922).CT Phase Ib28Safe, well tolerated, promising results in AML—68% of patients achieved CR [118].CT Phase I/II 24Safe, well tolerated, not sufficiently efficacious to recommend further development of the regimen in relapse or refractory CLL—33% of patients achieved PR but noone achieved CR [119].24Safe, well tolerated, moderately effective in relapsed ALL, and promising response in T-ALL—ORR was 33%, response was noted in 5 of 10 heavily pretreated T-ALL patients [89].27Safe, well tolerated, no therapeutic effect in AML [120].Data unpublishedData unpublished (NCT00093639).CT Phase II22Safe, well tolerated, modest efficacy in relapse or refractory CLL—4 of 22 patients with CLL achieved a PR [121].SirolimusmTOR inhibitor CT Phase I51Safe, well tolerated, promising results in AML—the ORR was 47% [92].12Combination of Decitabine and Sirolimus was safe and well tolerated. The primary focus of this Phase I study was not on measuring efficacy. However, it is worth noting that after one cycle, most patients showed stability or a positive response in their disease status [122].Data unpublishedData unpublished (NCT00068302, NCT00874562).CT Phase II5Poorly tolerated, no therapeutic effect in refractory or relapsed ALL (NCT01162551).Data unpublishedData unpublished (NCT00235560).CT Phase II26Safe, well tolerated, survival rates appear comparable to other salvage regimens in AML. The CR was 33%; median overall survival was 7.7 months in newly diagnosed elderly AML patients and 6.6 months in relapsed/refractory AML patients [123]. Data unpublishedData unpublished (NCT00776373).TemsirolimusmTOR inhibitor CT Phase I16Safe, well tolerated, promising results in relapsed or refractory childhood ALL—sixteen patients were included in the study, achieving an ORR of 47% and a CR of 27% [124].15Temsirolimus in combination with UK R3 chemotherapy can induce responses in children with ALL—this regimen induced remission in seven of fifteen patients with relapse ALL. However, this intensive regimen is associated with unacceptable toxicity [94,125].Data unpublishedData unpublished (NCT00101088).CT Phase II53Acceptable safety profile, promising results in AML—in 53 evaluable patients, the ORR was 21%. Median disease-free survival was 3–5 months, and median overall survival was 4 months [95].89Safe, well tolerated, underperforming results in CLL—one of 15 patients with CLL had PR [126].Data unpublishedData unpublished (NCT00084916, NCT00086840).Dual PI3K/mTORGedatolisibDual PI3K/mTOR inhibitorCT Phase II10Safe, well tolerated, no clinical benefit in relapse or refractor AML—no objective response was detected for any of the 10 patients [127].BEZ235Dual PI3K/mTOR inhibitorCT Phase I24Safe, well tolerated, clinical benefit for small subset of patients with ALL, with no benefit in patients with AML. CR observed in 3 of 24 patients, all of them ALL (3/11) [102].AktGSK2141795Akt inhibitorCT Phase II24Safe, well tolerated, no clinical benefit in AML with RAS mutations—no patient obtained CR. The study was closed early due to lack of clinical activity [113].


## 5. Summary and Perspective

Blood malignancies are highly heterogenous as they arise from different types of blood cells at distinct levels of differentiation. Notwithstanding the many differences related to origin and pathogenesis, all leukemia we described are characterized by disrupted PI3K/Akt/mTOR signaling pathway. Consequently, the PI3K/Akt/mTOR axis supports various hallmarks of cancer, including sustaining proliferative signaling, evading growth suppressors, activating invasion and metastasis, and deregulating cellular energetics. Furthermore, the pathway’s role in promoting resistance to traditional therapeutic approaches like chemotherapy and immunotherapy underscores its significance in disease progression. While advancements have been made in understanding the role of the PI3K/Akt/mTOR pathway in cancer, there is still much to explore. The complexity of its interactions with other cellular processes, its isoform-specific effects, and its crosstalk with various signaling pathways present both challenges and opportunities for targeted therapeutic interventions. Continued research into this pathway’s intricate mechanisms will likely yield novel insights and contribute to the development of more effective and personalized treatments for cancer, including hematological neoplasms, ultimately improving patients’ outcomes. Duvelisib and Idelalisib are the only PI3K/Akt/mTOR pathway inhibitors approved by the FDA for the treatment of relapsed CLL/SLL and FL at this point. Both of them are applied in resistant or relapsed CLL. Buparlisib shows efficacy in treating CLL but has significant side effects. Further research is therefore needed to increase the safety of such therapy while retaining the therapeutic effect. Umbralisib in combination with Ibrutinib shows promising results in the treatment of relapsed/refractory CLL. However, due to the limited number of studies on this topic, new research on efficacy of the drug is required. The abovementioned study on Everolimus application in AML shows its ineffectiveness against this disease. However, Everolimus combined with chemotherapy in pediatric ALL shows efficacy, although further confirmation of these data is necessary. Gedatolisib shows effectiveness in Ph+ ALL and AML in in vivo models. Nevertheless, to date, there are no clinical trials on the drug’s efficacy in AML treatment. Given the promising results of in vivo studies, the new clinical trials should be considered. On the other hand, BEZ235 as a dual PI3K/mTOR inhibitor shows limited efficacy in clinical trials in BCP-ALL and T-ALL. However, due to the very small patient population studied in Phase I clinical trials, expanded studies are required to make a clear conclusion. Due to the lack of significant clinical activity observed in the Phase II study of GSK2141795 in combination with Trametinib for relapsed/refractory AML patients with *RAS* mutation, and the early closure of the study, it has been suggested that this treatment approach may not be effective for this specific patient population. To sum up, inhibition of the PI3K/Akt/mTOR signaling pathway constitutes a promising treatment option not only for CLL but also for other blood malignancies, and this topic should be further addressed by future studies. We look forward to the future with hope that, one day, the application of these drugs will be expanded beyond CLL treatment and will bring relief to more hematological patients.

## Figures and Tables

**Figure 1 cancers-15-05297-f001:**
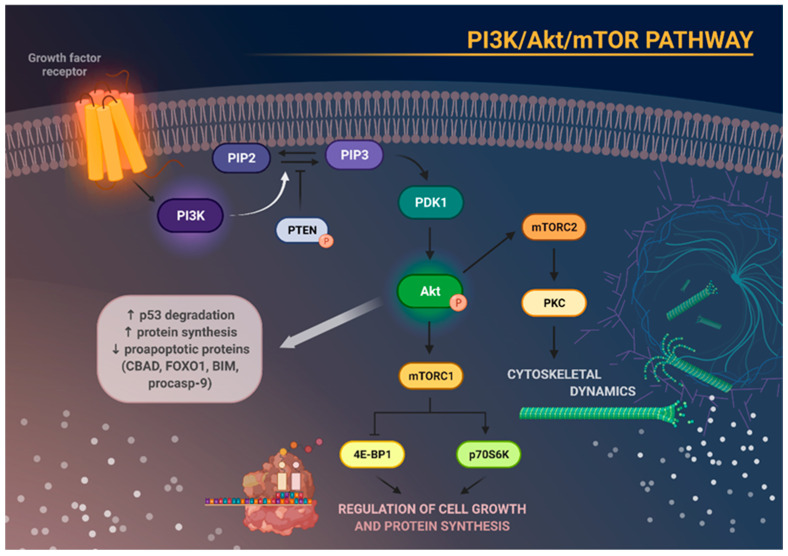
Schematic representation of the main molecular consequences of phosphoinositide 3-kinase/Akt/mammalian target of rapamycin (PI3K/Akt/mTOR) pathway activation. Stimulation of growth factor receptor activates PI3K, which in turn causes phosphorylation of phosphatidylinositol 4,5-bisphosphate (PIP2) to phosphatidylinositol 3,4,5-triphosphate (PIP3). This process can be reversed by phosphatase and tensin homolog (PTEN). Increased activity of PIP3 results in recruitment of phosphoinositide-dependent kinase 1 (PDK1), which subsequently phosphorylates and activates Akt. Activation of Akt degrades p53, increases protein synthesis, and inhibits the activity of several proapoptotic proteins. Akt, through activation of mTOR, enables the formation of the two complexes—mTORC1 and mTORC2. mTORC1 phosphorylates and activates eukaryotic translation initiation factor 4E-binding protein 1 (4E-BP1) and p70S6 kinase (p70S6K), which are responsible for cell growth and protein synthesis. mTORC2, on the other hand, regulates cytoskeletal architecture through protein kinase C (PKC) phosphorylation.

**Table 1 cancers-15-05297-t001:** The table presents the fundamental classes of the phosphatidylinositol-3 kinase (PI3K), along with their subclasses and isoforms. The table also includes genes encoding a given PI3K class.

PI3K Class	Subunits	Isoforms	Encoding Gene
IA	Catalytic subunits + regulating subunit isoform	p110α, p110β, p110δ	*PIK3CA*, *PIK3CB, PI3KCD*
IB	-	p110γ	*PIK3CG*
II	Only the catalytic subunit	PI3K-C2α, PI3K-C2β, PI3K-C2γ	*PIK3C2A, PIK3C2B, PIK3C2G*
III	Vps34	-	*PIK3C3*

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
