# Peer review of "PI3K/Akt/mTOR Signaling Pathway in Blood Malignancies—New Therapeutic Possibilities"

_cancers, 2023, doi:10.3390/cancers15215297_

Round 1

Reviewer 1 Report

Comments and Suggestions for Authors

In this review article, Wiese et al. summarize the PI3K/Akt/mTOR signaling pattern in leukemic cells. They also examine the various classes of inhibitors targeting this signaling network and the clinical trials which have been carried out in leukemic patients.

The manuscript is well written and quite comprehensive. However, enthusiasm for this kind of work is dampened by the fact that, apart from PI3K inhibitors in CLL, all the other class of drugs displayed no significant activity when tested in leukemic patients. I’m therefore asking what novelty such a review will bring to the field (see the sentence at lines 26-28 of the first page: “Recently, new inhibitors of PI3K/Akt/mTOR signaling have been developed…..”). None of the inhibitors highlighted here is new, everything which has been summarized in the review is well known.

Moreover, the Authors omitted to cite the only new class of mTOR inhibitors (although these drugs were first reported in 2016, so they too aren’t brand new), that is third-generation bi-steric inhibitors that are selective for mTOR complex-1 (mTORC1) (see Front Oncol. 2023 Apr 14;13:1162694. doi: 10.3389/fonc.2023.1162694. eCollection 2023.; Front Oncol. 2021 Aug 2;11:673213. doi: 10.3389/fonc.2021.673213. eCollection 2021.).

This class of drugs has not entered clinical trials for leukemias yet, although one of them is currently being tested in solid tumors (see NCT04774952). In any case, they would be worth to be cited, as they seem to be quite effective in pre-clinical models of Ph+ B-ALL.

Comments on the Quality of English Language

English style is fine , it will only require minor changes

Reviewer 2 Report

Comments and Suggestions for Authors

This is a comprehensive summary of the signalling pathway and therapeutics to target it. It is information dense and could benefit from more use of figures and tables to summarise key information, for example information on subclasses and dimerization of the receptor. Table 1 is useful but is not mentioned in the text – its existence should be flagged early in the therapeutics section.

When discussing therapeutics:

Number of patients in trials and responding should be reported in consistently – it is mentioned for some but not others in the table and response rates are reported in different ways for different studies

There is a brief mention of Venetoclax resistance – this assumes the reader knows what it targets and the resistance pathway. A brief context to standard therapeutics and where the targeted pathway inhibitors fit would be useful for non-specialist readers.

There are some errors in terminology/writing which undermine credibility in places. Specifically:

Section 5.2- ATP-competent should read ATP-competitive throughout

Line 383 – the study terminated due to excess mortality not the lack of it

Line 416 meaning of ‘grade 3 or 4 infection rate’ unclear

Table 1: Clear Responses to BEZ235 – does this mean complete?

Some minor copyediting is required– missing or unnecessary punctuation and some strikethroughs and underscores, minor writing errors and formatting of the table.

Reviewer 3 Report

Comments and Suggestions for Authors

This paper tries to summarize the new possibilities of targeting PI3K/Akt/mTOR Signaling Pathway in Leukemia Cells.

I would consider this paper only after major revision for the following reason

1- The Introduction to the pathway is simply too long and has been the topic of many excellent reviews.

2- The main introduction should focus on the Dysregulation of the PI3K/Akt/mTOR Pathway in Leukemia

3- Leukemia is a broad term for cancers of the blood cells. It would make a lot of sense therefore also to include not only AML/CML/PML but also Lymphomas (which the author do by citing T-ALL) where mainly these inhibitors have been tested clinically. To focus only on  AML/CML/MNPs etc. gives too little to review

-   - Also a Table with clinically approved PI3K/Akt/mTOR and those currently in clinical trials and preclinical stage would be great. Please be aware that there are now mutant selective PI3Ka inhibitors are being developed.

5- Unclear to me what are the new “Therapeutic Possibilities in Leukemia by targeting the PI3K/Akt/mTOR Signaling Pathway

Reviewer 4 Report

Comments and Suggestions for Authors

The review by Wiese et al. concerns the recent developments and therapeutic possibilities of molecules that act on the PI3K/Akt/mTOR pathway.

The manuscript is comprehensive and covers in detail the published papers over the last five years. Furthermore, the manuscript is very well written and easy to follow.

I recommend its publication in Cancers in its current version.

Comments on the Quality of English Language

The quality of English language is good, and in my opinion do not require further modification.

Round 2

Reviewer 1 Report

Comments and Suggestions for Authors

I have not changed my mind about this manuscript. The Authors have made some changes, however the topic itself is dead.